# One2All: Individual Reweighting for User-Oriented Fairness in Recommender Systems

## Abstract

Recommender systems often manifest biases toward a small user group, resulting in pronounced disparities in recommendation performance, i.e., the User-Oriented Fairness (UOF) issue. Existing research on UOF faces three major limitations, and no single approach effectively addresses all of them. **Limitation 1:** Post-processing methods fail to address the root cause of the UOF issue. **Limitation 2:** Some in-processing methods rely heavily on unstable user similarity calculations under severe data sparsity problems. **Limitation 3:** Other in-processing methods overlook the disparate treatment of individual users within user groups. In this paper, we propose a novel **I**ndividual **R**eweighting for **U**ser-**O**riented **F**airness framework, namely IR-UOF, to address all the aforementioned limitations. IR-UOF serves as a versatile solution applicable across various backbone recommendation models to achieve UOF. The motivation behind IR-UOF is to *introduce an in-processing strategy that addresses the UOF issue at the individual level without the need to explore user similarities.* We conduct extensive experiments on three real-world datasets using four backbone recommendation models to demonstrate the effectiveness of IR-UOF in mitigating UOF and improving recommendation fairness. The code of this paper is available at https://anonymous.4open.science/r/IR-UOF-D53B/

## 1 Introduction

Fairness is currently a critical research field in Recommender Systems (RSs) Deldjoo et al. (2022); Chen et al. (2023a). RS is a complex domain involving frequent interactions between users and items Zheng et al. (2022); Li et al. (2022), leading to fairness issues arising from both the user Li et al. (2021); Rahmani et al. (2022) and item side Dash et al. (2021); Deldjoo et al. (2021b). In this paper, we focus on the fairness issue related to performance disparities among different user groups.

RSs often exhibit bias toward a small group of users, resulting in significant unfairness in the quality of recommendations Li et al. (2021); Rahmani et al. (2022); Wen et al. (2022b), referred to as the **U**ser-**O**riented **F**airness (UOF) issue. We define users who receive more satisfying recommendation results as **advantaged users** and other users as **disadvantaged users**, following Li et al. (2021); Rahmani et al. (2022). Existing research has shown that advantaged users constitute only a small proportion of the total user base Li et al. (2021), as many users suffer from the data sparsity problem Han et al. (2023b) and fail to receive satisfactory recommendations. Therefore, addressing the UOF issue is crucial in RSs to enhance the overall quality of recommendation services.

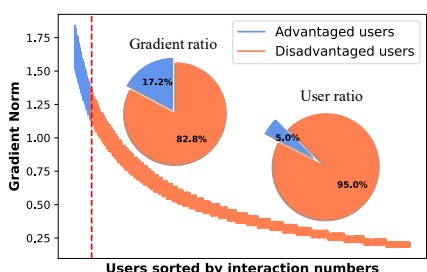

Figure 1: Gradients come from different users in a training epoch of LightGCN in the Epinion dataset.

Existing research in addressing the UOF issue includes post-processing methods (directly adjusting the recommendation lists) and in-processing methods (adjusting the training process of recommendations). All these methods face three key limitations, and *none of the existing research can address all of them.* **Limitation 1**: *Post-processing methods fail to address the root cause of the UOF issue.* Some

existing research proposes post-processing methods Li et al. (2021); Rahmani et al. (2022) to re-rank the calculated recommendation lists *after model training* to balance advantaged and disadvantaged user groups. However, *the root cause of the UOF issue lies in the unfair training process*, where the recommendation models are dominated by advantaged users Han et al. (2023a; 2024a). As illustrated in Figure 1, the advantaged users, who make up only 5% of the population, contribute 17.2% of the gradients in a single training epoch. Post-processing methods cannot mitigate the unfair training process and are thus unable to address the root cause of the UOF issue, resulting in limited performance.

**Limitation 2**: *Some in-processing methods rely heavily on unstable user similarity calculations under severe data sparsity problems.* Some studies Han et al. (2023a; 2024a;b) adopt in-processing methods to mitigate unfair training processes in recommendation models. These approaches calculate user similarities based on user-item interactions. Then they aim to enhance the training process for disadvantaged users by enabling them to learn from other similar users. However, disadvantaged users often face severe data sparsity problems Li et al. (2021); Rahmani et al. (2022), and the sparse interactions result in unstable similarity calculations based on user-item interactions. Consequently, the performance of these methods is limited.

**Limitation 3**: *Other in-processing methods overlook the disparate treatment of individual users within user groups.* Some methods Wen et al. (2022b) propose strategies that enhance the importance of loss values for the entire disadvantaged user group during model training. However, as illustrated in Figure 2, different users within disadvantaged or advantaged users also tend to be treated differently, i.e., the under-representation problem Chai & Wang (2022). The lack of individualized optimization strategies for each user reduces the effectiveness of these methods, resulting in less significant performance improvement Han et al. (2024a). Overall, *none of the existing research fully addresses all three of these limitations, leaving the UOF issue insufficiently resolved.*

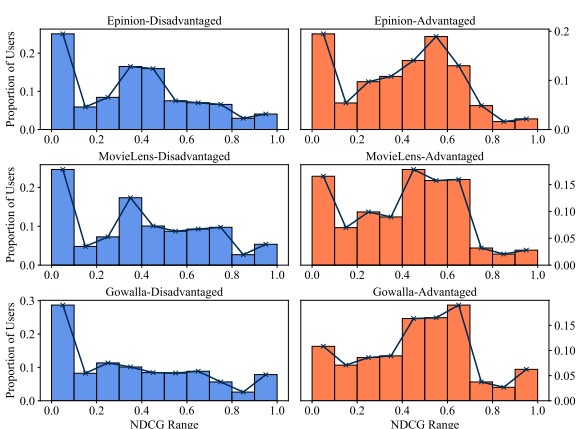

Figure 2: Distribution of user performance among advantaged and disadvantaged user groups, with LightGCN as the recommendation model.

In this paper, we propose a novel **I**ndividual **R**eweighting for **U**ser-**O**riented **F**airness framework, named IR-UOF, to address all the aforementioned limitations. IR-UOF serves as a versatile solution applicable across various backbone recommendation models to achieve UOF. The motivation behind IR-UOF is to *introduce an in-processing strategy that addresses UOF issues at the individual level without the need to explore user similarities based on user-item interactions.* In detail, to tackle **Limitation 1**, IR-UOF addresses the root cause of the UOF issue by introducing an in-processing strategy to focus on mitigating the unfair training processes of recommendation models. To tackle **Limitation 2**, IR-UOF avoids relying on user similarity calculations based on limited user-item interactions. Instead, IR-UOF focuses on the loss value of each user, as it directly reflects the quality of the user's training. By adjusting the weights of losses for different users, the model's emphasis on various users can be modulated. To address **Limitation 3**, IR-UOF introduces an individual-level reweighting method, transcending the constraints of group-level optimization. As illustrated in Figure2, it is not only disadvantaged users but also individual users among the advantaged group who may be overlooked. IR-UOF adaptively adjusts the weight of the loss value for all users, providing precise strategies tailored to each user's training situation. This approach allows IR-UOF to enhance the training quality of every individual user who is dominated by recommendation models, achieving overall fairness through individual optimization, i.e., One to All.

We conduct extensive experiments on three publicly available real-world datasets using four backbone recommendation models. The effectiveness of IR-UOF is comprehensively assessed using evaluation metrics from various perspectives. Experimental results demonstrate that IR-UOF outperforms all State-Of-The-Art (SOTA) methods across all datasets and backbone models.

We summarize our main contributions as follows: (1) We introduce an in-processing method to address the UOF issue. (2) We propose an individual reweighting strategy to achieve overall fairness by balancing the training process for different users. (3) We conduct extensive experiments to demonstrate the effectiveness of the proposed method.

## 2 RELATED WORK

### 2.1 FAIRNESS IN MACHINE LEARNING

As outlined in Mehrabi et al. (2021), fairness in decision-making processes is broadly defined as the absence of bias or favoritism toward an individual or group based on their inherent or acquired characteristics. Research on machine learning fairness can be categorized from various perspectives into distinct domains.

**Groups Affected by Fairness Issues.** Fairness research primarily addresses two aspects: group fairness and individual fairness Mehrabi et al. (2021); Dai et al. (2022). Group fairness aims to ensure equal treatment for users from different demographic groups. Key approaches in this domain include Demographic Parity Kusner et al. (2017), Equalized Odds Hardt et al. (2016), Equal Opportunity Hardt et al. (2016), Conditional Statistical Parity Corbett-Davies et al. (2017), and Treatment Equality Berk et al. (2021). Individual fairness focuses on providing similar recommendations to similar individuals. Research areas include Fairness Through Unawareness Grgic-Hlaca et al. (2016), Fairness Through Awareness Dwork et al. (2012), and Counterfactual Fairness Kusner et al. (2017).

**Stages of the ML Process.** Research on fairness can also be categorized according to different stages of the machine learning process: pre-processing methods, in-processing methods, and post-processing methods Mehrabi et al. (2021); Dai et al. (2022). Pre-processing methods aim to transform training data to eliminate underlying biases before model training d'Alessandro et al. (2017); Kang et al. (2020). In-processing methods incorporate fairness considerations into the training process to mitigate bias during model development Dai & Wang (2020); Bose & Hamilton (2019). Post-processing methods adjust the predictions of a trained model to ensure fairness Li et al. (2021).

In this paper, we introduce an in-processing framework specifically designed to ensure user-oriented fairness, a kind of group fairness in RSs.

### 2.2 FAIRNESS IN RECOMMENDER SYSTEMS

Fairness in RSs can be examined from three primary perspectives: user fairness, item fairness, and provider fairness Deldjoo et al. (2023).

**User Fairness:** Studies in this area aim to ensure that similar users receive comparable recommendation outcomes. Key considerations include ranking accuracy Deldjoo et al. (2021c), diversity coverage Melchiorre et al. (2021), under-ranking Gorantla et al. (2021), and selection rate Sühr et al. (2021). **Item Fairness:** The goal is to ensure that similar items receive equal exposure, regardless of sensitive attributes Rastegarpanah et al. (2019); Deldjoo et al. (2021a); Dash et al. (2021) or previous exposure history Biega et al. (2018), such as in cold-start scenarios. **Provider Fairness:** There is a tendency for providers with a more extensive interaction history to be recommended more frequently, creating a "superstar effect" Ferraro (2019); Gharahighehi et al. (2021). Efforts to mitigate exposure disparities arising from the relationship between providers and items Sühr et al. (2021) and private characteristics Shakespeare et al. (2020) are crucial for fostering an equitable market.

In this paper, we focus on the underexplored issue of user-oriented fairness (UOF), specifically addressing fairness among users with varying levels of activity.

## 3 PROBLEM FORMULATION

This paper focuses on achieving fairness in RSs through individual reweighting. We categorize the problem into two main components: user-oriented fairness and individual reweighting.

### 3.1 USER-ORIENTED FAIRNESS

In RSs, let $\mathcal{U}$ and $\mathcal{I}$ represent the user set and the item set, respectively. $N_{\mathcal{U}}$ and $N_{\mathcal{I}}$ represent the number of users and items. Following prior research Li et al. (2021); Rahmani et al. (2022); Han et al. (2023a), users are divided into two groups: disadvantaged users ($\mathcal{D}$) and advantaged users ($\mathcal{A}$). The disadvantaged group $\mathcal{D} = \{D_1, D_2, \ldots, D_{N_{\mathcal{D}}}\}$ consists of users with fewer interactions, while the advantaged group $\mathcal{A} = \{A_1, A_2, \ldots, A_{N_{\mathcal{A}}}\}$ includes users with more frequent interactions. Here, $N_{\mathcal{D}}$ and $N_{\mathcal{A}}$ represent the number of users in each group. Users with more interactions are more likely to be advantaged. The goal is to minimize the performance disparity between $\mathcal{D}$ and $\mathcal{A}$, thereby achieving UOF while maintaining overall recommendation quality.

UOF is a type of group fairness Hardt et al. (2016); Dwork et al. (2012), which ensures that groups of users with different protected attributes are treated comparably. Specifically, UOF aims to provide users with different activity levels with the same recommendation performance. The definition of UOF is as follows Li et al. (2021); Rahmani et al. (2022); Han et al. (2023a):

**Definition 1** (User-Oriented Fairness (UOF)).

$$\mathbb{E}[\mathcal{M}(\mathcal{A})] = \mathbb{E}[\mathcal{M}(\mathcal{D})]. \tag{1}$$

Here, $\mathcal{M}$ is a metric (e.g., NDCG and Hit Ratio) evaluating recommendation performance. $\mathcal{M}(u)$ represents the recommendation performance for user $u$.

UOF aims to offer users with different activity levels the same recommendation performance, which is always impossible in real-world RSs. Therefore, researchers Li et al. (2021); Rahmani et al. (2022); Han et al. (2023a) calculate the difference in average recommendation performance for different user groups to evaluate the fairness of a model:

**Definition 2** (The UOF metric ($\mathcal{M}_{UOF}$)).

$$\mathcal{M}_{UOF}(\mathcal{D}, \mathcal{A}) = \left| \frac{1}{|\mathcal{A}|} \sum_{A_i \in \mathcal{A}} \mathcal{M}(A_i) - \frac{1}{|\mathcal{D}|} \sum_{D_i \in \mathcal{D}} \mathcal{M}(D_i) \right|. \tag{2}$$

The $\mathcal{M}_{UOF}$ value is used to evaluate the fairness of a recommendation model. A lower $\mathcal{M}_{UOF}$ indicates a fairer algorithm, aiming for equal treatment across different user activity groups. Note that some researchers Han et al. (2024a) have proposed alternative metrics for assessing UOF. However, these metrics often replicate the characteristics of standard metrics like NDCG or Hit Ratio, and thus, they fail to specifically address the fairness aspect. Therefore, we utilize the more widely accepted $\mathcal{M}_{UOF}$ metric, as adopted by Li et al. (2021); Rahmani et al. (2022); Han et al. (2023a), which is better suited to evaluate fairness within RSs.

### 3.2 INDIVIDUAL REWEIGHTING

Individual reweighting aims to allocate different weights to each user's loss values, thereby giving more weight to users who are likely to be disadvantaged by the recommendation models. Consider the utility loss $\mathcal{L} = \sum_{U_i \in \mathcal{U}} L(U_i)$ of a recommendation model, where $L(U_i)$ represents the loss value for user $U_i$. For instance, this could be the cross-entropy loss. The individual reweighting strategy involves calculating a set of weights $\boldsymbol{\beta} = \{\beta_1, \beta_2, \ldots, \beta_{N_{\mathcal{U}}}\}$ that maximize the total weighted loss: $\max_{\boldsymbol{\beta}} \mathcal{L} = \sum_{U_i \in \mathcal{U}} \beta_i L(U_i)$. By amplifying the loss values of poorly trained users, this approach ensures that the model pays more attention to these users, thereby enhancing their training process and improving overall fairness.

## 4 METHODOLOGY

In this paper, we propose a novel **I**ndividual **R**eweighting for **U**ser-**O**riented **F**airness framework, named IR-UOF, to address the UOF issue in RSs. Designed as a versatile framework, IR-UOF can be integrated with any existing backbone recommendation model to enhance fairness. The key motivation behind IR-UOF is to introduce an in-processing framework that employs an individual-level optimization strategy, thereby avoiding the need to calculate user similarities in sparse datasets and overcoming the three key limitations. *Firstly*, IR-UOF assigns reweighting ratios to advantaged

and disadvantaged user groups, thereby tailoring reweighting strengths based on the training qualities of these groups. *Secondly*, IR-UOF provides a detailed calculation strategy for the individual reweighting process within each user group. *Thirdly*, IR-UOF introduces a delayed updating strategy to ensure the smooth optimization of the algorithm.

## 4.1 REWEIGHTING RATIOS FOR DIFFERENT USER GROUPS

As discussed in Section 1, users in both advantaged and disadvantaged groups can be adversely affected by recommendation models. Therefore, reweighting both user groups is essential to enhance overall fairness and improve recommendation performance. For the advantaged user group $\mathcal{A}$ and the disadvantaged user group $\mathcal{D}$, IR-UOF computes two weighting sets: $\boldsymbol{\beta}^{\mathcal{A}} = \{\beta_1^{\mathcal{A}}, \beta_2^{\mathcal{A}}, \ldots, \beta_{N_{\mathcal{A}}}^{\mathcal{A}}\}$ and $\boldsymbol{\beta}^{\mathcal{D}} = \{\beta_1^{\mathcal{D}}, \beta_2^{\mathcal{D}}, \ldots, \beta_{N_{\mathcal{D}}}^{\mathcal{D}}\}$. To ensure fairness and control the range of the loss function, these weights must be non-negative and capped. Hence, the weighting sets for these two groups must satisfy the following conditions:

$$\sum_{\beta_i^{\mathcal{A}} \in \boldsymbol{\beta}^{\mathcal{A}}} \beta_i^{\mathcal{A}} = K_{\mathcal{A}}, \beta_i^{\mathcal{A}} \geq 0; \quad \sum_{\beta_i^{\mathcal{D}} \in \boldsymbol{\beta}^{\mathcal{D}}} \beta_i^{\mathcal{D}} = K_{\mathcal{D}}, \beta_i^{\mathcal{D}} \geq 0. \tag{3}$$

Here, $K_{\mathcal{A}}$ and $K_{\mathcal{D}}$ control the reweighting strengths for each user group. To prioritize reweighting in user groups that are more likely to be dominated by recommendation models, we calculate these values as follows:

$$K_{\mathcal{A}} = \frac{\sum_{A_i \in \mathcal{A}} L(A_i)}{\sum_{A_i \in \mathcal{A}} L(A_i) + \sum_{D_i \in \mathcal{D}} L(D_i)} K; \quad K_{\mathcal{D}} = \frac{\sum_{D_i \in \mathcal{D}} L(D_i)}{\sum_{A_i \in \mathcal{A}} L(A_i) + \sum_{D_i \in \mathcal{D}} L(D_i)} K, \tag{4}$$

where $K$ is a hyperparameter that controls the overall reweighting scale. By calculating the reweighting ratios for different user groups, IR-UOF assigns greater weights to the user group which is more likely to be overlooked, ensuring balanced attention and enhanced fairness.

## 4.2 CALCULATION OF INDIVIDUAL RWEIGHTING STRATEGY

Since the calculation of individual reweighting strategy is the same for advantaged and disadvantaged user groups, we take the disadvantaged user group as an example in this section. The individual reweighting problem for disadvantaged users can be formulated as follows:

$$\max_{\boldsymbol{\beta}^{\mathcal{D}}} \mathcal{L}_{\mathcal{D}} = \sum_{D_i \in \mathcal{D}} \beta_i^{\mathcal{D}} L(D_i), \text{ s.t. } \sum_{\beta_i^{\mathcal{D}} \in \boldsymbol{\beta}^{\mathcal{D}}} \beta_i^{\mathcal{D}} = K_{\mathcal{D}}, \beta_i^{\mathcal{D}} \geq 0. \tag{5}$$

Naturally, the optimal solution of $\boldsymbol{\beta}^{\mathcal{D}*}$ in Problem equation 5 is assigning 1 to the largest loss and assigning 0 to all others. However, in order to tackle the UOF issue, consideration should be given to individual users more likely to be neglected, not just the single most likely. Hence, we introduce a regularization term and receive the following individual reweighting problem:

$$\max_{\boldsymbol{\beta}^{\mathcal{D}}} \mathcal{L}_{\mathcal{D}} = \sum_{D_i \in \mathcal{D}} \beta_i^{\mathcal{D}} L(D_i) - \alpha ||\boldsymbol{\beta}^{\mathcal{D}}||^2, \text{s.t.} \sum_{\beta_i^{\mathcal{D}} \in \boldsymbol{\beta}^{\mathcal{D}}} \beta_i^{\mathcal{D}} = K_{\mathcal{D}}, \beta_i^{\mathcal{D}} \geq 0. \tag{6}$$

A higher value of the hyperparameter $\alpha$ results in more positive weights. When $\alpha$ reaches a sufficiently large value, all samples will be assigned equal weights, and equation 6 will degenerate into the original recommendation model training process.

Directly calculating the optimal value of $\boldsymbol{\beta}^{\mathcal{D}}$ through the optimization process is time consuming and is not suitable for real practice. Therefore, we introduce the closed-form solution of adaptive weights $\boldsymbol{\beta}^{\mathcal{D}}$. Due to space limitations, the detailed proof and calculation process are provided in **Appendix A**.

Assume the loss of each disadvantaged user is represented as $l_i = L(D_i), D_i \in \mathcal{D}$. *Firstly*, sort the losses of all disadvantaged users in descending order, i.e., $l_i \geq l_j, \forall i > j$. *Secondly*, for $i$, calculate the value $\gamma$ following $\sum_{j=1}^{\gamma} l_j - \gamma l_{\gamma+1} > 2\alpha K_{\mathcal{D}} > \sum_{j=1}^{\gamma} l_j - \gamma l_\gamma$. Then, the optimal solution $\boldsymbol{\beta}^{\mathcal{D}*}$ of Problem equation 6 is as follows:

$$\beta_i^{\mathcal{D}*} = \text{ReLU}(\frac{\gamma l_i - \sum_{j=1}^{\gamma} l_j + 2\alpha K_{\mathcal{D}}}{2\alpha\gamma}). \tag{7}$$

We can follow a similar calculation process to get $\boldsymbol{\beta}^{\mathcal{A}*}$.

---

**Algorithm 1:** IR-UOF

---

**Input** : User set $\mathcal{U}$ and item set $\mathcal{I}$; Advantaged user set $\mathcal{A}$ and Disadvantaged user set $\mathcal{D}$;
Hyperparameters $K$ and $\alpha$; Training round limit $T$; $t = 0$.
**Output** : Final Recommention model;

**1 while** $t < T$ **do**

**2**     Extract loss values $\{L(A_1), L(A_2), \ldots, L(A_{N_\mathcal{A}})\}$ and $\{L(D_1), L(D_2), \ldots, L(D_{N_\mathcal{D}})\}$;

**3**     Calculate $\beta^{\mathcal{A}(t)*}$ and $\beta^{\mathcal{D}(t)*}$ according to problem equation 6;

**4**     Get the value of $\beta^{\mathcal{A}(t)}$ and $\beta^{\mathcal{D}(t)}$ according to Equation equation 8;

**5**     Calculate the final loss $L_{fairness}$ and do a gradient updation.

**6 end**

**7 Return** trained recommendation model;

---

### 4.3 Delayed Updating Strategy for Individual Reweighting

For such an individual reweighting method, directly replacing the value of $\beta^\mathcal{A}$ and $\beta^\mathcal{D}$ in each training round with the optimal one may result in unsteadiness. Therefore, we introduce a learning rate schedule to update $\beta^\mathcal{A}$ and $\beta^\mathcal{D}$ smoothly in the training round $t$:

$$\beta^{\mathcal{A}(t)} = (1 - \eta_t)\beta^{\mathcal{A}(t-1)} + \eta_t\beta^{\mathcal{A}(t)*}, \beta^{\mathcal{D}(t)} = (1 - \eta_t)\beta^{\mathcal{D}(t-1)} + \eta_t\beta^{\mathcal{D}(t)*}, \tag{8}$$

where $\eta^t = 1 - \frac{t}{T}$, with $T$ denoting the total training round. By doing so, IR-UOF avoids drastic changes to the individual weights and brings a more stable training process.

From the above discussion, we can find that both $\alpha$ and $K$ ($K_\mathcal{A}$ and $K_\mathcal{D}$ are calculated from $K$) have an impact on the values of $\beta^\mathcal{A}$ and $\beta^\mathcal{D}$. We decide the values of these two hyperparameters according to the experimental results in Section 5.6.

### 4.4 In-processing Training Strategy

As an in-processing framework, IR-UOF can be integrated with any existing backbone recommendation model to enhance fairness. In the training round $t$ of a given recommendation model, IR-UOF *firstly* extracts original training losses of advantaged users $\{L(A_1), L(A_2), \ldots, L(A_{N_\mathcal{A}})\}$ and disadvantaged users $\{L(D_1), L(D_2), \ldots, L(D_{N_\mathcal{D}})\}$. *Secondly*, IR-UOF calculates the optimal individual reweighting sets $\beta^{\mathcal{A}(t)*}$ and $\beta^{\mathcal{D}(t)*}$ based on these loss values according to problem equation 6. *Thirdly*, IR-UOF utilizes the delayed updating strategy to get the value of $\beta^{\mathcal{A}(t)}$ and $\beta^{\mathcal{D}(t)}$ in $t$-th training round. *Finally*, IR-UOF aggregates the final loss function with the fairness concern as follows:

$$L_{fairness} = \sum_{A_i \in \mathcal{A}} \beta_i^{\mathcal{A}(t)} L(A_i) + \sum_{D_i \in \mathcal{D}} \beta_i^{\mathcal{D}(t)} L(D_i). \tag{9}$$

This loss function is used for the gradient update of the recommendation model. *To avoid overfitting to noisy user-item training pairs, the reweighting strategy is applied at the user level rather than at the individual training sample level.* We outlined the overall algorithm of IR-UOF in algorithm 1.

## 5 Experiments and Analysis

To comprehensively evaluate the effectiveness of the proposed IR-UOF framework, we conduct extensive experiments on three real-world datasets to address the following Research Questions (RQs): **RQ1:** How does IR-UOF compare with existing SOTA methods in tackling the UOF issue and improving overall recommendation performance? **RQ2:** What is the impact of reweighting different user groups on the performance of IR-UOF? **RQ3:** As an in-processing method, is IR-UOF time-efficient? **RQ4:** Can IR-UOF maintain satisfactory performance in extremely sparse datasets? **RQ5:** How do important hyperparameters affect the performance of IR-UOF? **RQ6:** How robust is the generalizability of the IR-UOF framework when faced with variations in the classification of advantaged and disadvantaged users?

## 5.1 DATASETS AND EXPERIMENTAL SETTINGS

Due to space limitations, the details of this section are provided in **Appendix B**.

**Dataset Description.** We utilize three real-world datasets: **Epinion** Massa & Avesani (2007), **MovieLens** Harper & Konstan (2015), and **Gowalla** Liu et al. (2017), which are commonly used to validate the UOF issueRahmani et al. (2022); Han et al. (2023a).

**Baselines and Backbone Models.** We compare IR-UOF with the SOTA UOF methods UFR (failing to tackle Limitation 1) Li et al. (2021), In-UCDS Han et al. (2023a), HyperUOF Han et al. (2024b), and II-GOOT Han et al. (2024a) (failing to tackle Limitation 2), S-DRO Wen et al. (2022a) (failing to tackle Limitation 3). Besides, we choose four different backbone recommendation models, including a traditional matrix factorization method (MF Koren et al. (2009)), two deep-learning-based methods (NeuMF He et al. (2017), VAECF Liang et al. (2018)), and a graph neural network-based method (LightGCN He et al. (2020)).

**Evaluation Protocols.** *(1) User Grouping:* Users are ranked by interaction counts, with the top 5% as advantaged and the rest as disadvantaged Li et al. (2021); Dai et al. (2022); Han et al. (2023a). *(2) Performance Metrics:* We adopt the widely-used Normalized Discounted Cumulative Gain (NDCG) Wang et al. (2013) and Hit Ratio (HR) Waters (1976) to evaluate the recommendation performance of each model. Besides, we utilize $\mathcal{M}_{UOF}$ to evaluate the UOF level of a recommendation model, with a lower value means a fairer performance. *(3) Statistical Robustness:* Each evaluation is repeated 10 times, reporting average performance with significance testing (p-value < 0.05).

**Parameter Settings.** The details of this part are provided in **Appendix B**.

## 5.2 OVERALL COMPARISON (RQ1)

To comprehensively evaluate the effectiveness of the proposed IR-UOF framework, we conduct extensive experiments on three publicly available real-world datasets using four backbone models. The results are detailed in Table 1. Across all datasets, IR-UOF consistently outperforms all SOTA methods, highlighting the importance of addressing all three critical limitations (introduced in Section 1) in solving UOF.

**(1) Tackling Limitation 1 (Compared with UFR).** IR-UOF outperforms UFR across all datasets. Unlike the post-processing method UFR, IR-UOF effectively mitigates the training gap between advantaged and disadvantaged users, thereby addressing Limitation 1. UFR's failure to tackle the root cause of the UOF issue results in its generally poor performance compared with all baselines.

**(2) Tackling Limitation 2 (Compared with In-UCDS, HyperUOF, and II-GOOT).** IR-UOF consistently outperforms In-UCDS, HyperUOF, and II-GOOT across all datasets, especially on the sparser Gowalla dataset. These methods rely heavily on identifying essential similarities among users based on user-item interactions, a process that is unstable under severe data sparsity. As a result, their performance is limited, particularly in sparser datasets. IR-UOF focuses on reweighting loss values for each user, directly reflecting their training quality, making it a more stable and direct solution to the UOF issue.

**(3) Tackling Limitation 3 (Compared with S-DRO).** Compared to S-DRO, IR-UOF achieves fairer models with better recommendation performance. S-DRO sorely assign more weight to the entire disadvantaged user group, overlooking the different treatments individual users within this group receive. This inability to target users individually limits its performance. IR-UOF adaptively reweights each user's loss value, providing unique treatment to each user and accurately improving the training of users dominated by recommendation models. Thus, IR-UOF can narrow the training gap across the entire user group, naturally improving both fairness and recommendation performance.

## 5.3 ABLATION STUDY (RQ2)

We conduct an ablation study to analyze the impact of reweighting different user groups on the performance of IR-UOF, using LightGCN as the backbone model. As illustrated in Table 2, "Re-Adv" indicates reweighting only in the advantaged user group (setting $K_{\mathcal{A}} = K$), while "Re-Dis" indicates reweighting only in the disadvantaged user group (setting $K_{\mathcal{D}} = K$).

Table 1: Overall experimental result.

| | | | Epinion | | | | MovieLens | | | | Gowalla | | | |
|---|---|---|---|---|---|---|---|---|---|---|---|---|---|---|
| | | | Over. | Adv. | Dis. | $\mathcal{M}_{UOF}$ | Over. | Adv. | Dis. | $\mathcal{M}_{UOF}$ | Over. | Adv. | Dis. | $\mathcal{M}_{UOF}$ |
| MF | NDCG | Original | 0.361 | 0.387* | 0.359 | 0.028 | 0.394 | 0.453* | 0.391 | 0.062 | 0.356 | 0.458* | 0.350 | 0.108 |
| | | S-DRO | 0.364 | 0.376 | 0.364 | 0.012 | 0.396 | 0.431 | 0.394 | 0.037 | 0.357 | 0.433 | 0.352 | 0.080 |
| | | UFR | 0.362 | 0.382 | 0.361 | 0.021 | 0.399 | 0.429 | 0.397 | 0.032 | 0.361 | 0.440 | 0.357 | 0.083 |
| | | In-UCDS | 0.368 | 0.381 | 0.367 | 0.014 | 0.410 | 0.439 | 0.408 | 0.030 | 0.357 | 0.433 | 0.353 | 0.080 |
| | | HyperUOF | 0.366 | 0.380 | 0.365 | 0.015 | 0.403 | 0.440 | 0.401 | 0.038 | 0.360 | 0.434 | 0.356 | 0.078 |
| | | II-GOOT | 0.370 | 0.380 | 0.370 | 0.010 | 0.404 | 0.443 | 0.402 | 0.040 | 0.361 | 0.437 | 0.357 | 0.080 |
| | | **IR-UOF** | **0.377*** | **0.382** | **0.377*** | **0.005*** | **0.412*** | **0.433** | **0.411*** | **0.022*** | **0.373*** | **0.440** | **0.369*** | **0.071*** |
| | HR | Original | 0.460 | 0.513* | 0.457 | 0.057 | 0.474 | 0.548* | 0.470 | 0.077 | 0.432 | 0.593* | 0.424 | 0.169 |
| | | S-DRO | 0.462 | 0.511 | 0.460 | 0.051 | 0.475 | 0.531 | 0.472 | 0.059 | 0.437 | 0.572 | 0.430 | 0.142 |
| | | UFR | 0.460 | 0.513* | 0.457 | 0.056 | 0.472 | 0.521 | 0.470 | 0.052 | 0.447 | 0.581 | 0.440 | 0.142 |
| | | In-UCDS | 0.462 | 0.507 | 0.460 | 0.047 | 0.485 | 0.530 | 0.483 | 0.047 | 0.447 | 0.571 | 0.440 | 0.131 |
| | | HyperUOF | 0.462 | 0.510 | 0.460 | 0.050 | 0.484 | 0.530 | 0.482 | 0.048 | 0.443 | 0.570 | 0.436 | 0.134 |
| | | II-GOOT | 0.465 | 0.509 | 0.463 | 0.046 | 0.487 | 0.532 | 0.485 | 0.047 | 0.442 | 0.573 | 0.435 | 0.138 |
| | | **IR-UOF** | **0.478*** | **0.509** | **0.477*** | **0.033*** | **0.489*** | **0.531** | **0.486*** | **0.044*** | **0.464*** | **0.584** | **0.458*** | **0.126*** |
| NeuMF | NDCG | Original | 0.369 | 0.393* | 0.368 | 0.025 | 0.404 | 0.484* | 0.400 | 0.084 | 0.376 | 0.467* | 0.371 | 0.096 |
| | | S-DRO | 0.373 | 0.386 | 0.372 | 0.014 | 0.398 | 0.465 | 0.394 | 0.071 | 0.376 | 0.459 | 0.372 | 0.086 |
| | | UFR | 0.371 | 0.390 | 0.370 | 0.020 | 0.403 | 0.480 | 0.399 | 0.081 | 0.380 | 0.461 | 0.376 | 0.085 |
| | | In-UCDS | 0.374 | 0.389 | 0.374 | 0.015 | 0.408 | 0.476 | 0.405 | 0.072 | 0.384 | 0.460 | 0.381 | 0.080 |
| | | HyperUOF | 0.371 | 0.390 | 0.370 | 0.020 | 0.407 | 0.479 | 0.403 | 0.076 | 0.384 | 0.459 | 0.380 | 0.079 |
| | | II-GOOT | 0.376 | 0.390 | 0.376 | 0.014 | 0.414 | 0.480 | 0.410 | 0.070 | 0.383 | 0.459 | 0.379 | 0.081 |
| | | **IR-UOF** | **0.381*** | **0.392** | **0.380*** | **0.012*** | **0.423*** | **0.480** | **0.420*** | **0.060*** | **0.401*** | **0.460** | **0.398*** | **0.062*** |
| | HR | Original | 0.472 | 0.522* | 0.469 | 0.053 | 0.481 | 0.576* | 0.476 | 0.101 | 0.441 | 0.601* | 0.433 | 0.168 |
| | | S-DRO | 0.470 | 0.515 | 0.468 | 0.046 | 0.484 | 0.569 | 0.479 | 0.090 | 0.442 | 0.586 | 0.434 | 0.151 |
| | | UFR | 0.471 | 0.510 | 0.469 | 0.041 | 0.476 | 0.569 | 0.471 | 0.098 | 0.441 | 0.583 | 0.434 | 0.150 |
| | | In-UCDS | 0.482 | 0.515 | 0.480 | 0.036 | 0.505 | 0.571 | 0.502 | 0.069 | 0.446 | 0.587 | 0.438 | 0.149 |
| | | HyperUOF | 0.476 | 0.514 | 0.474 | 0.040 | 0.504 | 0.571 | 0.500 | 0.071 | 0.445 | 0.589 | 0.437 | 0.152 |
| | | II-GOOT | 0.475 | 0.513 | 0.473 | 0.041 | 0.507 | 0.573 | 0.504 | 0.069 | 0.443 | 0.591 | 0.435 | 0.156 |
| | | **IR-UOF** | **0.490*** | **0.518** | **0.489*** | **0.029*** | **0.514*** | **0.572** | **0.511*** | **0.061*** | **0.470*** | **0.594** | **0.463*** | **0.131*** |
| VAECF | NDCG | Original | 0.374 | 0.405* | 0.372 | 0.033 | 0.416 | 0.492 | 0.412 | 0.080 | 0.399 | 0.463* | 0.396 | 0.067 |
| | | S-DRO | 0.374 | 0.397 | 0.373 | 0.024 | 0.421 | 0.496* | 0.417 | 0.079 | 0.413 | 0.455 | 0.410 | 0.045 |
| | | UFR | 0.372 | 0.400 | 0.370 | 0.030 | 0.416 | 0.485 | 0.412 | 0.073 | 0.399 | 0.454 | 0.396 | 0.058 |
| | | In-UCDS | 0.382 | 0.398 | 0.381 | 0.016 | 0.433 | 0.490 | 0.429 | 0.060 | 0.406 | 0.454 | 0.404 | 0.051 |
| | | HyperUOF | 0.381 | 0.398 | 0.380 | 0.018 | 0.432 | 0.489 | 0.429 | 0.060 | 0.403 | 0.455 | 0.400 | 0.054 |
| | | II-GOOT | 0.385 | 0.400 | 0.384 | 0.015 | 0.434 | 0.491 | 0.430 | 0.060 | 0.402 | 0.456 | 0.399 | 0.057 |
| | | **IR-UOF** | **0.391*** | **0.400** | **0.391*** | **0.010*** | **0.435*** | **0.489** | **0.432*** | **0.057*** | **0.423*** | **0.458** | **0.421*** | **0.037*** |
| | HR | Original | 0.475 | 0.527* | 0.472 | 0.055 | 0.460 | 0.569* | 0.454 | 0.114 | 0.427 | 0.594* | 0.419 | 0.175 |
| | | S-DRO | 0.475 | 0.519 | 0.473 | 0.046 | 0.469 | 0.555 | 0.465 | 0.090 | 0.444 | 0.583 | 0.437 | 0.146 |
| | | UFR | 0.476 | 0.524 | 0.473 | 0.051 | 0.463 | 0.560 | 0.458 | 0.101 | 0.431 | 0.584 | 0.423 | 0.161 |
| | | In-UCDS | 0.487 | 0.521 | 0.485 | 0.036 | 0.478 | 0.559 | 0.473 | 0.086 | 0.433 | 0.581 | 0.425 | 0.156 |
| | | HyperUOF | 0.489 | 0.520 | 0.487 | 0.033 | 0.479 | 0.558 | 0.475 | 0.083 | 0.432 | 0.580 | 0.424 | 0.156 |
| | | II-GOOT | 0.491 | 0.520 | 0.489 | 0.031 | 0.481 | 0.561 | 0.476 | 0.085 | 0.428 | 0.584 | 0.420 | 0.164 |
| | | **IR-UOF** | **0.498*** | **0.524** | **0.496*** | **0.028*** | **0.486*** | **0.564** | **0.482*** | **0.082*** | **0.453*** | **0.590** | **0.446*** | **0.144*** |
| LightGCN | NDCG | Original | 0.399 | 0.440 | 0.397 | 0.043 | 0.483 | 0.529 | 0.480 | 0.049 | 0.403 | 0.486* | 0.399 | 0.087 |
| | | S-DRO | 0.401 | 0.438 | 0.399 | 0.039 | 0.496 | 0.532* | 0.494 | 0.038 | 0.417 | 0.483 | 0.414 | 0.069 |
| | | UFR | 0.400 | 0.440 | 0.398 | 0.042 | 0.488 | 0.523 | 0.486 | 0.037 | 0.407 | 0.480 | 0.403 | 0.077 |
| | | In-UCDS | 0.406 | 0.436 | 0.405 | 0.031 | 0.501 | 0.528 | 0.499 | 0.028 | 0.418 | 0.484 | 0.414 | 0.070 |
| | | HyperUOF | 0.406 | 0.438 | 0.404 | 0.034 | 0.501 | 0.527 | 0.500 | 0.027 | 0.406 | 0.480 | 0.402 | 0.078 |
| | | II-GOOT | 0.408 | 0.437 | 0.407 | 0.030 | 0.504 | 0.529 | 0.502 | 0.026 | 0.409 | 0.482 | 0.406 | 0.077 |
| | | **IR-UOF** | **0.416*** | **0.441*** | **0.414*** | **0.027*** | **0.515*** | **0.529** | **0.515*** | **0.014*** | **0.423*** | **0.484** | **0.420*** | **0.064*** |
| | HR | Original | 0.479 | 0.559 | 0.475 | 0.084 | 0.549 | 0.619 | 0.545 | 0.074 | 0.480 | 0.604* | 0.473 | 0.131 |
| | | S-DRO | 0.483 | 0.553 | 0.479 | 0.074 | 0.558 | 0.617 | 0.554 | 0.062 | 0.486 | 0.596 | 0.480 | 0.116 |
| | | UFR | 0.480 | 0.553 | 0.476 | 0.077 | 0.550 | 0.615 | 0.546 | 0.069 | 0.486 | 0.598 | 0.480 | 0.118 |
| | | In-UCDS | 0.484 | 0.554 | 0.480 | 0.074 | 0.566 | 0.620 | 0.563 | 0.057 | 0.482 | 0.593 | 0.476 | 0.116 |
| | | HyperUOF | 0.485 | 0.555 | 0.481 | 0.074 | 0.555 | 0.614 | 0.552 | 0.062 | 0.484 | 0.597 | 0.478 | 0.119 |
| | | II-GOOT | 0.488 | 0.557 | 0.484 | 0.073 | 0.569 | 0.619 | 0.566 | 0.053 | 0.481 | 0.595 | 0.475 | 0.119 |
| | | **IR-UOF** | **0.502*** | **0.560*** | **0.499*** | **0.061*** | **0.582*** | **0.621*** | **0.580*** | **0.040*** | **0.503*** | **0.602** | **0.498*** | **0.104*** |

\* Over. indicates overall performance. Adv. and Dis. indicate the performance of advantaged and disadvantaged users, respectively.

\* The results of IR-UOF are highlighted in bold. The best results are marked with *. The second-best results are underlined.

\* All outcomes pass the significance test, with a p-value below the significance threshold of 0.05.

Overall, IR-UOF achieves the best performance compared to the ablation methods, demonstrating that reweighting both advantaged and disadvantaged user groups is necessary to improve fairness and recommendation performance. This is because some users in both groups may receive poor training results. Reweighting only advantaged or disadvantaged users neglects some poorly trained users. Since disadvantaged users are more likely to be dominated by recommendation models, Re-Dis achieves significantly better performance than Re-Adv.

Table 2: Ablation study.

| | | Epinion | | MovieLens | | Gowalla | |
|---|---|---|---|---|---|---|---|
| | | Overall | $\mathcal{M}_{UOF}$ | Overall | $\mathcal{M}_{UOF}$ | Overall | $\mathcal{M}_{UOF}$ |
| NDCG | Original | 0.399 | 0.043 | 0.483 | 0.049 | 0.403 | 0.087 |
| | Re-Adv | 0.402 | 0.046 | 0.492 | 0.058 | 0.413 | 0.095 |
| | Re-Dis | 0.411 | 0.032 | 0.510 | 0.021 | 0.419 | 0.069 |
| | IR-UOF | **0.416** | **0.027** | **0.515** | **0.014** | **0.423** | **0.064** |
| HR | Original | 0.479 | 0.084 | 0.549 | 0.074 | 0.480 | 0.131 |
| | Re-adv | 0.483 | 0.099 | 0.552 | 0.082 | 0.486 | 0.152 |
| | Re-dis | 0.500 | 0.069 | 0.571 | 0.054 | 0.491 | 0.114 |
| | IR-UOF | **0.502** | **0.061** | **0.582** | **0.040** | **0.503** | **0.104** |

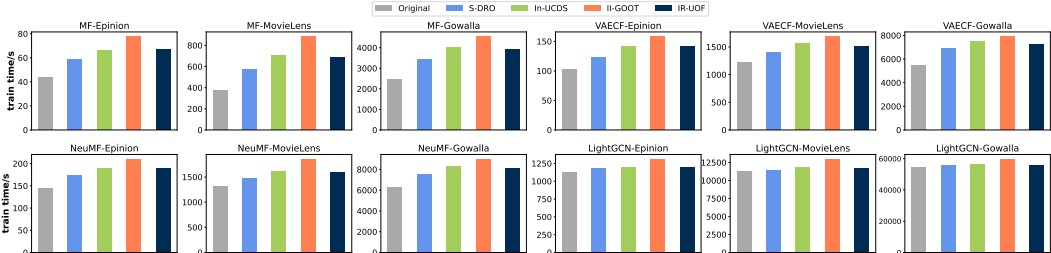

Figure 3: Training time of the original backbone model, S-DRO, In-UCDS, II-GOOT, and IR-UOF in datasets Epinion, MovieLens, and Gowalla.

## 5.4 MODEL EFFICIENCY (RQ3)

This section aims to analyze the efficiency of IR-UOF. As an in-processing framework, it is crucial to demonstrate that it does not impose excessive additional time costs on the original models' training process. We compare the training time of IR-UOF with all in-processing methods. The experimental results are presented in Figure 3. *Our analysis reveals that IR-UOF maintains a time cost comparable to the original backbone model, incurring only a slight additional time cost.* The additional time expenditure associated with IR-UOF is primarily due to sorting the loss values of users, which has a time complexity of $\mathcal{O}(N_{\mathcal{U}} \log N_{\mathcal{U}})$. This sorting process is time-efficient and depends solely on the size of the datasets. Consequently, when applied to more complex and time-intensive backbone models (e.g., LightGCN), the relative increase in training time due to IR-UOF becomes less significant. This observation underscores IR-UOF's applicability and practicality, especially in scenarios where backbone models are inherently resource-intensive.

## 5.5 SPARSE TEST (RQ4)

This section aims to demonstrate the superior performance of IR-UOF on extremely sparse datasets, thus highlighting the necessity of addressing **Limitation 2**. Disadvantaged users frequently encounter severe data sparsity issues. The capability of a model to handle sparse datasets is crucial for effectively tackling the UOF challenge. To validate this, we simulate sparser environments by *randomly omitting various ratios of interactions* within each dataset and subsequently conducting experiments. As outlined in Section 1, the user similarity calculation processes in In-UCDS, HyperUOF, and II-GOOT become unstable in sparse datasets, limiting these methods' performance. Therefore, we compare the performance of IR-UOF with these meth-

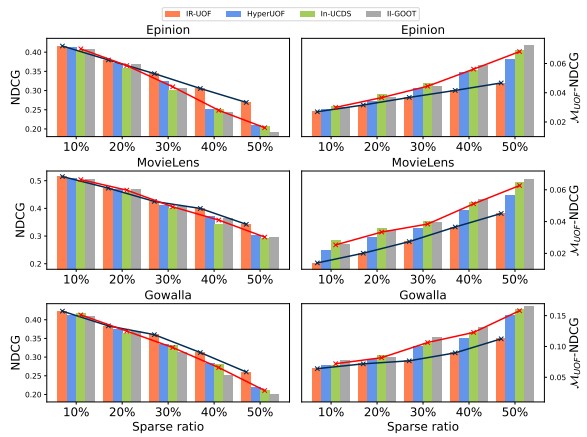

Figure 4: Sparse test with LightGCN as the backbone model.

ods. The experimental results are provided in Figure 4, with the "Sparse ratio" representing the percentage of interactions omitted.

The experimental findings unequivocally indicate that *IR-UOF consistently surpasses In-UCDS, HyperUOF, and II-GOOT across all levels of dataset sparsity, achieving superior recommendation performance and fairness.* Notably, the advantage of IR-UOF becomes more pronounced as the sparsity of the datasets increases. This is because, as datasets become sparser, the unstable user similarity calculation process for In-UCDS, HyperUOF, and II-GOOT significantly hampers their performance, leading to a faster rate of performance decline. In contrast, IR-UOF focuses on users' loss values, which can consistently and accurately reflect the user's training level, thereby making its performance more robust.

### 5.6 EFFECT OF HYPERPARAMETERS (RQ5)

In this section, we conduct experiments to analyze the effects of hyperparameters $\alpha$ and $K$ within the proposed IR-UOF framework. Due to space constraints, we present the experimental results using LightGCN as the backbone model in Figure 5.

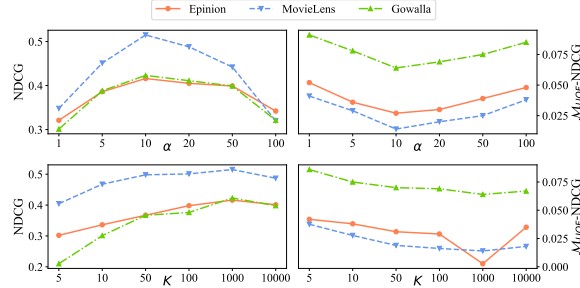

**Effect of $\alpha$.** As depicted in Figure 5, the IR-UOF framework achieves peak performance in terms of recommendation quality and UOF optimization when $\alpha$ is set to 10. The parameter $\alpha$ controls the importance

Figure 5: Effect of hyperparameters.

of the regularization term. A value of $\alpha$ that is too small will result in IR-UOF assigning positive weights to only a few users, thereby neglecting too many individuals. Conversely, a value of $\alpha$ that is too large will cause IR-UOF to degenerate into average weighting.

**Effect of $K$.** Figure 5 shows that IR-UOF attains optimal recommendation performance and UOF optimization when $K$ is set to 1000. This parameter controls the scale of weights assigned to each individual user in IR-UOF. A larger value of $K$ can better represent the differences in loss values among users, making recommendation models focus more on users who are often neglected. However, an excessively large value of $K$ causes the recommendation models to focus too much on specific users, thereby reducing the model's generalization ability.

## 6 GENERALIZABILITY OF IR-UOF (RQ6)

The level of data sparsity varies across different datasets, leading to a dynamic nature in how advantaged and disadvantaged users are categorized. To prove the generalizability of IR-UOF, we adopt LightGCN as the backbone model and modify the percentage of advantaged users from 5% to 50% within the Gowalla Dataset. The outcomes are presented in Figure 6. The experimental results indicate that the ratio of advantaged to disadvantaged users does not significantly affect the overall model performance. This is because IR-UOF reweights samples within both user groups, rather than a specific group. As the threshold increases, the overall activity level difference between the two user groups decreases, resulting in a reduced disparity in UOF gap. The results

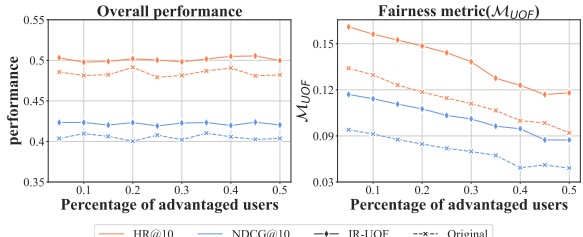

Figure 6: The above results illustrate how the overall performance and $\mathcal{M}_{UOF}$ of IR-UOF and the original model change in response to variations in the categorization of advantaged and disadvantaged users. Due to space limitations, we take LightGCN as the backbone model and Gowalla as the experimental dataset.

prove that IR-UOF has strong generalizability in narrowing the recommendation gap across various user distributions.

## 7 CONCLUSION

This paper addresses the User-Oriented Fairness (UOF) issue in Recommender Systems (RSs), specifically focusing on narrowing the recommendation gap between advantaged and disadvantaged user groups. We introduce a novel framework, **I**ndividual **R**eweighting for **U**ser-**O**riented **F**airness, referred to as IR-UOF, designed to overcome three significant limitations that existing research has not adequately addressed. IR-UOF adaptively reweights the loss values of each user, ensuring that no users are dominated by the recommendation models. As a result, IR-UOF enhances the training quality for each individual user, achieving overall fairness and improving recommendation performance through individualized optimization, i.e., One to All. We conduct extensive experiments on three real-world datasets to demonstrate the efficacy of IR-UOF.

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

## A  DETAILED PROOF AND CALCULATION OF INDIVIDUAL REWEIGHING STRATEGY

The optimization of $\beta^{\mathcal{D}}$ in problem equation 6 can be formulated as

$$\max_{\beta^{\mathcal{D}}} \mathcal{L}_{\mathcal{D}} = \sum_{D_i \in \mathcal{D}} L(D_i) - \alpha \|\beta^{\mathcal{D}}\|^2, \text{ s.t. } \sum_{\beta_i^{\mathcal{D}} \in \beta^{\mathcal{D}}} \beta_i^{\mathcal{D}} = K_{\mathcal{D}}, \beta_i^{\mathcal{D}} \geq 0.$$

Denote the loss of each sample as $l_i = L(D_i), D_i \in \mathcal{D}$, the optimization problem can be written as

$$\min_{\beta^{\mathcal{D}}} - \sum_i l_i \beta_i^{\mathcal{D}} + \alpha \|\beta^{\mathcal{D}}\|^2, \text{ s.t. } \sum_{\beta_i^{\mathcal{D}} \in \beta^{\mathcal{D}}} \beta_i^{\mathcal{D}} = K_{\mathcal{D}}, \beta_i^{\mathcal{D}} \geq 0. \tag{10}$$

Consider the Lagrangian dual of problem equation 10, we employ the KKT conditions Boyd & Vandenberghe (2004) to optimize equation 10 via its Lagrangian:

$$L(\beta^{\mathcal{D}}, \lambda, \nu) = -\sum_i l_i \beta_i^{\mathcal{D}} + \alpha \sum_i \beta_i^{\mathcal{D}2} - \sum_i \lambda_i \beta_i^{\mathcal{D}} + \nu(\sum_i \beta_i - K_{\mathcal{D}}). \tag{11}$$

From the KKT conditions, we have

$$\nabla_{\beta_i^{\mathcal{D}}} L(\beta^{\mathcal{D}}, \lambda, \nu) = -l_i + 2\alpha\beta_i^{\mathcal{D}} - \lambda_i + \nu = 0, \tag{15a}$$

$$\lambda_i \beta_i^{\mathcal{D}} = 0, \beta_i^{\mathcal{D}} \geq 0, \lambda_i \geq 0, i = [1, N_D], \tag{15b}$$

$$\sum_i \beta_i^{\mathcal{D}} = K_D. \tag{15c}$$

Combining equation 15a and equation 15b, since $\alpha > 0$, we can derive that

$$\begin{cases} l_i - \nu > 0 \Rightarrow 2\alpha\beta_i^{\mathcal{D}} = l_i - \nu, \lambda_i = 0, \\ l_i - \nu \leq 0 \Rightarrow 2\alpha\beta_i^{\mathcal{D}} = 0, \lambda_i = \nu - l_i. \end{cases} \tag{13}$$

Hence, we get $\beta_i^{\mathcal{D}} = (\frac{l_i - \nu}{2\alpha})_+ = \max\{\frac{l_i - \nu}{2\alpha}, 0\}$. Then equation 15c can be written as

$$\sum_i (l_i - \nu)_+ = 2\alpha K_{\mathcal{D}}. \tag{14}$$

Here, $\nu$ can be calculated through solving equation 14. Firstly, we consider that $\sum_i l_i \leq 2\alpha K_{\mathcal{D}}$, the unique solution is

$$\nu = (2\alpha K_{\mathcal{D}} - \sum_i l_i)/N_{\mathcal{D}}. \tag{15}$$

When $0 < 2\alpha K_{\mathcal{D}} < \sum_i l_i$, there exist $l_{\max} > \nu \geq 0$ such that $\sum_i (l_i - \nu)_+ = 2\alpha K_{\mathcal{D}}$ holds true. Without loss of generality, suppose that the loss vector $l$ is sorted in descending order, i.e., $l_{\max} = l_1 > l_2 > \cdots > l_{N_{\mathcal{D}}} = l_{\min}$, there exists a $\gamma \in [1, N_{\mathcal{D}}]$ that satisfies $l_\gamma > \nu > l_{\gamma+1}$. The problem equation 14 can be expressed as

$$\sum_i (l_i - \nu)_+ = \sum_{i=1}^{\gamma} l_i - \gamma\nu = 2\alpha K_{\mathcal{D}}, \tag{16}$$

$$\Rightarrow \nu = \frac{\sum_{i=1}^{\gamma} l_i - 2\alpha K_{\mathcal{D}}}{\gamma}.$$

Moreover, with the expression of $\nu$, it can be derived that the index $\gamma$ satisfies

$$\sum_{i=1}^{\gamma} l_i - \gamma l_{\gamma+1} > 2\alpha K_{\mathcal{D}} > \sum_{i=1}^{\gamma} l_i - \gamma l_\gamma. \tag{17}$$

Combining equation 15 and equation 16, the optimal solution of weight $\beta$ is

$$\beta_i^{\mathcal{D}*} = (\frac{l_i - (\sum_{i=1}^{\gamma} l_i - 2\alpha K_{\mathcal{D}})/\gamma}{2\alpha})_+ = \text{ReLU}(\frac{\gamma l_i - \sum_{j=1}^{\gamma} l_j + 2\alpha K_{\mathcal{D}}}{2\alpha\gamma}), \tag{18}$$

where $\gamma$ satisfies equation 17. When $\sum_i l_i \leq 2\alpha K_{\mathcal{D}}$, let $\gamma = N_{\mathcal{D}}$.

## B    DATASETS AND EXPERIMENTAL SETTINGS

**Dataset Description.** We utilize three publicly available real-world datasets: **Epinion** Massa & Avesani (2007), **MovieLens** Harper & Konstan (2015), and **Gowalla** Liu et al. (2017), representing different domains (opinions, movies, and points of interest). These datasets are commonly used to validate the performance of models addressing the UOF issueRahmani et al. (2022); Han et al. (2023a). Table 3 provides the statistics of these datasets.

Overall, we select these datasets for three specific reasons in order to demonstrate the scalability, efficiency, and effectiveness of IR-UOF. *(1) Domain Diversity:* The datasets originate from different domains, providing a comprehensive evaluation of IR-

Table 3: The statistics of datasets.

| Dataset | Users | Items | Interactions | Sparsity | Domain |
|---------|-------|-------|--------------|----------|--------|
| Epinion | 2,677 | 2,060 | 103,567 | 98.12% | Opinion |
| MovieLens | 5,738 | 3,627 | 760,814 | 96.34% | Movie |
| Gowalla | 33,699 | 123,587 | 1,011,694 | 99.98% | POI |

UOF's performance. *(2) Sparsity Variation:* The datasets vary in sparsity levels, which directly impacts the UOF issue due to differences in user activity. *(3) Scalability Testing:* The datasets differ in size, allowing us to demonstrate IR-UOF's scalability.

**Baselines and Backbone Models.** We compare IR-UOF with the SOTA UOF methods UFR (failing to tackle Limitation 1) Li et al. (2021), In-UCDS Han et al. (2023a), HyperUOF Han et al. (2024b), and II-GOOT Han et al. (2024a) (failing to tackle Limitation 2 and Limitation 3), S-DRO Wen et al. (2022a) (failing to tackle Limitation 3). (1) **UFR:** A post-processing re-ranking method that modifies recommendation results of a given backbone model. (2) **In-UCDS:** An in-processing method that allows disadvantaged users to learn from advantaged users based on dominant sets. (3) **HyperUOF:** An in-processing strategy that utilizes hypergraph to explore high-order correlations among advantaged and disadvantaged users. (4) **II-GOOT:** An in-processing framework that narrows the training gap between advantaged and disadvantaged users through intra- and inter-group stages. (5) **S-DRO** An in-processing method that minimizes the loss function value for disadvantaged users during training.

To fully evaluate the performance of IR-UOF and SOTA methods, we choose four different backbone recommendation models, including a traditional matrix factorization method (MF), two deep-learning-based methods (NeuMF, VAECF), and a graph neural network-based method (LightGCN). (1) **MF** Koren et al. (2009): Matrix Factorization maps both users and items to a joint latent factor space and calculates the similarities among users and items. (2) **NeuMF** He et al. (2017): Neural Collaborative Filtering introduces a deep neural network with non-linear activation functions to train a user and item matching function. (3) **VAECF** Liang et al. (2018): Variational Autoencoders for Collaborative Filtering proposes a generative model with multinomial likelihood and uses Bayesian inference for parameter estimation. (4) **LightGCN** He et al. (2020): Light Graph Convolution Network simplifies the design of GCN by including only the most essential component in GCN neighborhood aggregation — for collaborative filtering.

**Evaluation Protocols.** *(1) User Grouping:* Users are ranked by interaction counts, with the top 5% as advantaged and the rest as disadvantaged Li et al. (2021); Dai et al. (2022); Han et al. (2023a). *(2) Dataset Spilting:* Using the Leave-One-Out (LOO) strategy Chen et al. (2023b); He et al. (2017), we split data into training, validation, and testing sets. *(3) Performance Metrics:* We adopt the widely-used Normalized Discounted Cumulative Gain (NDCG) Wang et al. (2013) and Hit Ratio (HR) Waters (1976) to evaluate the recommendation performance of each model. A higher value indicates superior recommendation performance, with a predicted cut-off of $topK = 10$ Li et al. (2021); Dai et al. (2022); Han et al. (2023a). Besides, we utilize $\mathcal{M}_{UOF}$ to evaluate the UOF level of a recommendation model, with a lower value of $\mathcal{M}_{UOF}$ means a fairer performance. *(4) Statistical Robustness:* Each evaluation is repeated 10 times, reporting average performance with significance testing (p-value < 0.05).

**Parameter Settings.** *(1) For IR-UOF:* Hyperparameters $K$ and $\alpha$ are set according to Section 5.6. *(2) For UFR, In-UCDS, HyperUOF, and II-GOOT:* We use the code provided by authors and leave the parameters as their default values. *(3) For S-DRO:* Implemented as recommended in Wen et al. (2022b), with hidden layer dimensions (128, 64) and temperature $\tau$ set to 0.07. *(4) For backbone models:* We set the dimension of user and item embeddings to 64 for all of them, and adopt their

parameters as suggested in their original paper. We adopt the Adam optimizer Kingma & Ba (2014) with a learning rate of 0.0001 and ensure convergence through 200 training epochs for all models.

**Experiments Compute Resources.** We conducted our experiments on a GPU server equipped with 8 CPUs and an NVIDIA RTX 3090 (24G).

