# OpenReview forum: "One to All: Individual Reweighting for User-Oriented Fairness in Recommender Systems"
_ICLR.cc/2025/Conference — ICLR 2025 Conference Withdrawn Submission_

### Official Review · Reviewer_jTou · 2024-10-26

**Soundness:** 2
**Presentation:** 3
**Contribution:** 2
**Rating:** 3
**Confidence:** 5

**Summary:**

This paper proposed an individual re-weighting method IR-UOF to achieve user fairness by reweighting each indidual's weight during training. The main contirbution of this paper includes: (1) the authors have idnetified and summerized several limitations existing methods. (2) the authors streesed the importance of individual-level in-process methods.

**Strengths:**

- The authors comprehensively reviewed the limitations in previous works, using various representations such as figures and some preliminary results (Fig1 and Fig2).
- The experiments are complete and comprehensive, including ablation study and various analysis.

**Weaknesses:**

- The presentation of Section 4.1 and 4.2 could be improved. Authors only focus on how to presenting their framework while ignoring the motivation behind it. For example, it is hard to understand the intuition of Eq4 and the reason authors chose this formulation.

- The individual re-weighting methods to ensure UOF is not very novel. For example, a naive solution is to set the weight for each user as their individual popularity (levels of activity). Authors should discuss why this naive solution is not good. Additional, some existing papers, such as "Mitigating Popularity Bias for Users and Items with Fairness-centric Adaptive Recommendation" (https://doi.org/10.1145/3564286) also tackle the UOF issue from a individual perspective.

**Questions:**

- Can authors provide more intuitions when presneting the framework?

- Can authors include a more conprehensive literature review and discussion to existing methods including the mentioned naive solution?

---

### Official Review · Reviewer_nSMK · 2024-10-29

**Soundness:** 2
**Presentation:** 2
**Contribution:** 2
**Rating:** 5
**Confidence:** 4

**Summary:**

The paper attempts to address an important issue in recommendation systems, namely fairness. The authors first highlight the limitations of current methods and emphasize the need to focus on the underlying causes of unfairness, specifically that models often overemphasize users from advantaged groups with extensive interaction records. They propose Individual Reweighting for the User-Oriented Fairness method (IR-UOF) and conduct extensive experiments to validate the effectiveness of their method.

**Strengths:**

1. The authors aim to improve the recommendation accuracy of disadvantaged groups at the individual level during the training process.
2. To enable the model to focus more on disadvantaged group, the authors propose calculating loss weights for each user. To compute this more efficiently, they employ an approximate algorithm and provide relevant theoretical proof.
3. The experiments in the paper indicate that IR-UOF achieves good performance, improving both recommendation fairness and accuracy.

**Weaknesses:**

1. The paper's novelty is limited, as it merely assigns specific loss weights to each user to direct the model's focus toward disadvantaged groups. While this method may improve the prediction accuracy for some disadvantaged users, the fundamental issue remains that their low accuracy is primarily due to fewer interaction records. In my view, increasing weights does not address the root cause of User-Oriented Fairness (UOF), thus failing to resolve Limitation 1 identified by the authors.
2. The group fairness proposed by the authors is similar to group Distributionally Robust Optimization (DRO) in machine learning [1], so it is suggested to include related work on DRO and to add comparisons with some DRO models in the experiments.
3. There are some confusing aspects in the paper, such as in line 249, where the maximum loss weight is assigned a value of 1. Additionally, the authors do not explain why there may be individuals in the advantaged group who could also be overlooked by the model, given that these individuals have sufficient interaction records.
4. I agree that reweighting individuals in the advantaged group can improve their recommendation accuracy, but this may further exacerbate the differences between the two groups, leading to greater unfairness. However, based on the ablation study results presented in the paper (Table 2), reweighting the entire dominant group simultaneously improves both accuracy and fairness, which is a point of confusion for me.
5. My final concern is regarding the hyperparameter. As the authors mention in the paper (Line 257,500), this is inconsistent with the results shown in Figure 5. For example, when the NDCG in Figure 5 is approximately 0.32, while in Table 1, the NDCG is 0.483. This suggests that the conclusions drawn by the authors may be incorrect (When reaches a sufficiently large value, all samples will be assigned equal weights).
[1] Sagawa, Shiori, et al. "Distributionally Robust Neural Networks." International Conference on Learning Representations.

**Questions:**

See Weaknesses

---

### Official Review · Reviewer_dC7C · 2024-11-03

**Soundness:** 1
**Presentation:** 1
**Contribution:** 1
**Rating:** 1
**Confidence:** 5

**Summary:**

The paper presents a multimodal recommendation framework, MTSTRec, which utilizes a Time-aligned Shared Token (TST) fusion module to enhance cross-modal interactions in sequential recommendations. While the authors claim that their approach effectively aligns features from different modalities at each time step, the innovation appears limited, as many prior works have explored similar multimodal fusion strategies. Notably, the authors do not address the substantial latent space differences between modalities, particularly between item IDs and other data types, and they lack experimental evidence to substantiate their claims against state-of-the-art methods. Furthermore, while they suggest that MTSTRec can be applied beyond recommendations, no supporting experiments are provided. The baselines chosen for comparison are outdated, lacking the robustness of modern sequential recommendation algorithms. Overall, the motivation for the work is weak, and there is insufficient exploration of existing problems that could have been highlighted through preliminary studies.

**Strengths:**

The paper presents a structured framework for multimodal recommendations that integrates various data types.

**Weaknesses:**

Limited Novelty: The approach of aligning features across different modalities is not particularly innovative, given existing literature on multimodal fusion.

Lack of Addressing Latent Space Differences: The authors fail to consider the significant differences in latent spaces between modalities, which could affect the model’s performance.

Outdated Baselines: The choice of baselines for comparison lacks relevance, as they are not based on current sequential recommendation methods, weakening the impact of the findings.

Weak Motivation: The paper does not adequately emphasize the significance of the problem it addresses, lacking persuasive preliminary evidence.

Insufficient Insights: The limited experimental exploration reduces the potential contributions of the work to the community.

**Questions:**

How does the proposed method perform compared to well-known sequential recommendation models such as SASRec and BERT4Rec?

What differences in effectiveness can be observed between MTSTRec and other models focused on ID-based recommendations, multimodal recommendations, and pure modality-based recommenders?

How to validate the claims about the applicability of the method beyond recommendation tasks with relevant experiments?

---

### Official Review · Reviewer_y7GY · 2024-11-03

**Soundness:** 2
**Presentation:** 3
**Contribution:** 2
**Rating:** 3
**Confidence:** 3

**Summary:**

The authors propose a training algorithm for recommender models, aiming to ensure user-oriented fairness in prediction under given advantaged and disadvantaged user groups.
The proposed objective is based on the usual adversarial training, but the novelty is in its hierarchical (group and individual) reweighting strategy.
An empirical evaluation is extensively performed on three real-world datasets.
However, to my understanding, the proposed objective is a trivial combination of conventional methods based on adversarial training
while the current manuscript lacks discussion on the comparison with existing individual reweighting methods in recommender systems.

**Strengths:**

(S1) The empirical evaluation is extensively conducted.

(S2) The reproducible code is available.

**Weaknesses:**

### (W1) The novelty of a proposed method is unclear.
The proposed objective in Eq. (9) is a trivial combination of group-wise and individual reweighting.
Although the authors claim "The key motivation behind IR-UOF is to introduce an in-processing framework that employs an individual level optimization strategy" in line 213, there is no discussion on existing individual reweighting methods.
For example, Shivaswamy and Garcia-Garcia [b] have proposed an individual reweighting strategy for recommender systems based on adversarial training [a].
They introduced adversarial training to ensure user-oriented fairness without demographics (e.g., user group or similarity).
In addition, the formulation in Eq. (5) is quite similar to DRO. Togashi et al. [c] have proposed CVaR optimization (a dual of DRO)
and discussed smoothing for adversarial individual weights, which is closely related to the regularization technique in Section 4.2 of the current manuscript.

[a] Lahoti, Preethi, et al. "Fairness without demographics through adversarially reweighted learning." Advances in neural information processing systems 33 (2020): 728-740.

[b] Shivaswamy, Pannaga, and Dario Garcia-Garcia. "Adversary or friend? an adversarial approach to improving recommender systems." Proceedings of the 16th ACM Conference on Recommender Systems. 2022.

[c] Togashi, Riku, et al. "Safe Collaborative Filtering." The Twelfth International Conference on Learning Representations. 2024.

### (W2) The connection between the UOF metric in Eq. (2) and the proposed objective is unclear.
The authors introduced a UOF metric in Eq. (2) and then proposed to optimize Eq. (9).
However, the connection between Eq. (2) and Eq. (9) is non-trivial.
In the extreme case without any regularization proposed in Eq. (6), the sum of maximum loss values in advantaged/disadvantaged groups is minimized.
Since loss values should negatively correlate with metric values, the sum of minimum metric values in each user group is maximized.
In my opinion, there is no trivial reason to believe the difference of means (UOF) is minimized by maximizing minimums.
It would be helpful to provide a more rigorous theoretical analysis or justification for how optimizing Eq. (9) leads to minimizing the UOF metric in Eq. (2).

**Questions:**

### (Q1) On the computational efficiency of the proposed method
In Section 5.4, the authors claim that IR-UOR is applicable for complex and time-intensive backbone models (e.g., LightGCN).
To my understanding, Algorithm 1 requires the computation of all users' loss values to compute the optimal user weights **while fixing the model weights/parameters**.
It implies an extra inference cost for all user/item embeddings through the backbone model in every training step,
and this cost is much larger than the O(N log N) cost of sorting one-dimensional user loss values.
Therefore, the proposed algorithm is inapplicable to all NN-based models using stochastic gradient descent in real-world applications.
Any clarifications on this point would be helpful.

### (Q2) On the computation of user loss values
I noticed that the authors' implementation of LightGCN is based on pointwise binary cross-entropy, which is different from the original implementation of LightGCN (pairwise loss).
Since the pointwise loss function is not a "user-wise/listwise" loss,
the exact computation of L(u) requires exhaustive loss computation for all items for each user.
The pairwise loss also requires infeasible computation for all possible item pairs for each user.
So, the algorithm described in the current manuscript seems infeasible for the datasets (in particular, Gowalla).
Did the authors apply approximation for user loss computation (e.g., negative sampling) in the experiments?

If such an approximation technique is used,
the approximation error can affect the final performance, and
the robustness of the proposed method should be evaluated on a dataset with a large **item** set to show the generalizability of the proposed method.
Moreover, the authors should report the setting of mini-batch sizes, negative sampling strategies, and loss functions as well as learning rates (in line 810).

---

### Note · Authors · 2024-11-30

I have read and agree with the venue's withdrawal policy on behalf of myself and my co-authors.